# Influence of Catch Crops on Yield and Chemical Composition of Winter Garlic Grown for Bunch Harvesting

**Andrzej Sałata** [1] , **Gaetano Pandino** [2] , **Halina Buczkowska** [1,*] **and Sara Lombardo** [2]

1    Department of Vegetable Crops and Medicinal Plants, University of Life Sciences in Lublin, 20-950 Lublin, Poland; andrzej.salata@up.lublin.pl
2    Di3A- Dipartimento di Agricoltura, Alimentazione e Ambiente, University of Catania, 95123 Catania, Italy; g.pandino@unict.it (G.P.); sara.lombardo@unict.it (S.L.)
*    Correspondence: halina.buczkowska@up.lublin.pl; Tel.: +48-81-4456964

**Abstract:** The cultivation of catch crops left on the surface of the field in the form of mulch promotes sustainable farming practices, while protecting the biodiversity of agricultural landscape. The paper presents results of research from 2013–2016, aimed at determining the usefulness of catch crops of millet, buckwheat, white mustard, bird's-foot and Egyptian clover for soil mulching in winter garlic cultivation. The effect of soil litter on the amount of garlic crop in cultivation for bunch harvest and nutritional value determined by chemical composition of edible parts was determined. In the edible part of garlic, the content of dry matter, total and reducing sugars, L-ascorbic acid, total ash, crude fiber, phenolic acids and essential oil was evaluated. The control consisted of plots without mulch plants. In the cultivation of garlic under organic mulch, there was no decrease in commercial yield and no negative competitive effect on yielding. Garlic plants cultivated with mulch plants were characterized by increased height and developed more leaves. The catch crops used in the form of mulch did not affect the dry matter and total ash content in garlic bulbs. In the cultivation with plant litter, the concentration of phenolic acids and essential oil in the leaves was higher and the content of crude fiber was lower, compared to the cultivation without litter. Biomass from catch crops from clover and mustard increased the content of L-ascorbic acid, at the same time reducing the content of total and reducing sugars in the edible part of garlic, except for mulch plants of the bean family. The chemical composition of garlic was affected by different thermal and precipitation conditions in the years of research. In 2016, the year with the highest total rainfall, plants accumulated more dry matter, L-ascorbic acid as well as total and reducing sugars than in 2014 and 2015, years with less rainfall. Our research indicates that there are prospects for practical application of catch crops in the form of mulch for garlic cultivation to enhance the level of nutrients, without compromising the yield.

**Keywords:** *Allium sativum* L.; type of mulch; marketable yield; chemical composition; phenolic acids; essential oil

---

## 1. Introduction

In the world production, garlic (*Allium sativum* L.) ranks second after onion among cultivated bulbous vegetables. In Poland, consumers mostly use garlic bulbs, whereas the cultivation of garlic for an early bunch-harvest is still not very popular. However, at this stage, both bulbs and young leaves, which are rich in minerals and vitamins, are suitable for consumption. Garlic leaves can be consumed fresh or lyophilized and used in the food industry as an additive to meat and milk products. Garlic is one of the most versatile plant materials, with strong bactericidal and fungicidal, anti-atherosclerotic and general strengthening effects [1]. Its most important active component is an essential oil, whose main ingredients are allyl sulfides, phenolic acids, and L-ascorbic acid [2,3]. The quality of garlic bulbs

and leaves depends on their chemical composition, which is modified by genetic and environmental factors [4,5]. Among agrotechnical factors, it is organic fertilization which has a beneficial effect on garlic chemical composition [6]. The development of integrated and organic garlic cultivation encourages the use of catch crops, which can be a source of biomass replacing manure.

In commercial garlic production, intensive mineral fertilization is used and chemical weed control is carried out. Due to the weak root system, garlic has a low ability to uptake and absorb minerals from the soil. Most mineral soils require a high total dose of mineral fertilization (400–900 kg·ha$^{-1}$) [7]. Such a cultivation system contributes to chemical compounds run-off from the soil surface and leaching of nitrates in the autumn-winter period [8], thus, one of the basic assumptions about using catch crops for soil mulching is to reduce nitrogen loss in winter [9]. In years with high rainfall, N losses are usually 15–45 kg·ha$^{-1}$ [10]. Catch crops grown as mulch can reduce nitrogen leaching by up to 40%–50% [11], at the same time increasing the availability of nitrogen in succeeding crops [12]. Spring catch crops can be sown in summer and grown as companion planting with the main crop until they finish their cycle in winter. After winter, the biomass from catch crops covers the field with a layer of natural mulch and may remain there until the end of vegetation of the species grown as the main crop [13]. Cultivation of catch crops is an integral part of ecological and sustainable agriculture [14].

It is often emphasized in the literature that catch crops can be used (grown in the same place) for a number of years, and a favorable effect is observed for 4-5 consecutive years [15]. Based on the research, it was established that catch crops affect the soil in a multidirectional manner, primarily contributing to the formation and maintenance of the soil lump structure [16], shade the soil and reduce evapotranspiration [17], as well as improve its biological activity [18]. To date, there are divergent opinions on the effects of plant coverings on yielding. Beneficial crop-forming effect of organic mulch has been shown in the cultivation of garlic [19], sweet pepper [20], eggplant [21], broccoli [22], cabbage [23], carrot [24] and tomato [22,25,26]. There are also examples of lowering the yields in cultivation with green litter compared to the traditional cultivation of eggplant [27], zucchini [28], and celery [29]. Some authors associate the effect of lowering the crop with the occurrence of adverse weather conditions, as the reductions in question are small or absent in favorable weather conditions [14,30]. It seems that it is the choice of catch crop species which has a significant impact on the yield of the plant grown as the main crop [31–34]. It was found that catch crops with a broad C: N ratio [Paniculate (*Poaceae*), mustard: > 30] release small amounts of nitrogen, and as a result, they increase the amount of permanent humus components. In contrast, catch crops with narrow C: N ratios (bean: < 15) release a lot of nitrogen, but do not increase the amount of organic matter [35].

Currently, intensive searching for plant species that can be introduced into companion planting is underway [5]. The key issue in winter garlic cultivation is soil protection against both nitrate and erosion leaching in autumn and winter, as well as the increase of biological activity in spring [7]. In addition, garlic has high moisture requirements, thus there is a danger of competition with companion species.

The agricultural practice of growing winter garlic with catch crops is not yet recognized. Hence, there is a need to conduct research in this area. In this work, it was assumed that the biomass of catch crops left until spring in the form of mulch would not cause any reduction in the yield compared to conventional cultivation. The purpose of the research was also to determine the extent to which organic biomass can affect the chemical composition of edible parts of garlic in cultivation for bunch harvesting. In the edible part of garlic, the content of dry matter, total and reducing sugars, L-ascorbic acid, total ash, crude fiber, phenolic acids and essential oil was evaluated.

## 2. Materials and Methods

### 2.1. Location of Experimental Station and Climatic Conditions

Agrotechnical experiments were carried out in the years 2013–2016, in the town of Palikije (51.23° N; 22.31° E) at the Małopolska Plant Breeding Agency in Zamość. Chemical composition

determinations were made at the Department of Vegetable and Medicinal Plants, University of Life Sciences in Lublin.

The Lublin region stands out against the remaining area of Poland with a long retention of snow cover, the occurrence of large amounts of rainfall in the warm season, or long periods of no-rainfall. The average annual air temperature is from 7.6 to 8.0 °C, and during the growing season 13.2 °C. In Lublin region, the month with the lowest temperature is January, with an average temperature from −4.3 to −2.6 °C, while the highest temperatures are recorded in July (17.2–18.5 °C). A common phenomenon in this area is the large temperature variability in winter months. The number of hot days (with a maximum daily temperature of > 25 °C) is on average 28–37 days. The growing season lasts on average 215 days. The annual rainfall for the Lublin region is 550-600 mm. Annual rainfall is more prevalent in summer (June–August) over winter (December–February). Summer precipitation accounts for 35%–40% of the total amount. On a monthly basis, the highest amounts are recorded in July (above 75 mm), and the lowest in January and February (25–40 mm).

Meteorological data (i.e., total rainfall and average air temperatures) were monitored during each growing season in the years of study by a meteorological station sited Felin (51.13° N; 22.37° E).

## 2.2. Field Trials and Description of Cultivation

In the cultivation of winter garlic (cv. 'Arcus'), catch crops left after freezing as natural litter were: millet (*Panicum miliaceum* L.) 'Charkowskie 31', buckwheat (*Fagopyrum esculentum* L.) 'Kora', white mustard (*Sinapsis alba* L.) 'Borowska', bird's-foot (*Ornithopus sativus* Brot.) 'Emena' and Egyptian clover (*Trifolium alexandrinum* L.). The control consisted of plots without cover plants.

Millet, buckwheat, white mustard, bird's-foot and Egyptian clover were grown as spring catch crops in the years 2013–2016. In the 2nd decade of July, a field was prepared for catch crops to cover the soil surface; the soil liming was carried out. At the end of July, mineral fertilization was applied. The dose of phosphorus and potassium for all catch crops was (in kg·ha$^{-1}$): P$_2$O$_5$ 50 (in the form of triple superphosphate), K$_2$O 90 (in the form of potassium sulfate). Nitrogen fertilization was varied. A dose of N 20 kg·ha$^{-1}$ was used for bird's-foot and Egyptian clover, N 60 kg·ha$^{-1}$ (in the form of urea) for millet, buckwheat and white mustard. At the end of March (2014, 2015, 2016), additional N mineral fertilization (20 kg·ha$^{-1}$ as ammonium nitrate) was applied. No pesticides were used in the cultivation of garlic. Doses of phosphorus and potassium in the control plots were like those used in the plots with cover, and in spring, nitrogen fertilization was provided only.

The experiment was established as a single-factor one by means of completely randomized blocks in four replications. The area of each plot with a cover crop was 8.0 m$^2$ (2.0 × 4.0 m). Garlic cloves were planted in rows every 30 cm, every 2 cm in each row. Each combination had 160 garlic plants in individual replicate.

The main physico-chemical characteristics of soil (at the layer of 0–20 cm) were the following (expressed as %): sand 35.2, loam 25.8, clay 39, organic matter 1.6, pH in KCl 6.7, Ca 4.5, N total 0.68, P 1.2, K 1.8, Mg 0.9.

Cover plants were sown at the beginning of August (2013, 2014, 2015). After freezing, the mass of plants grown became mulch, which covered the soil surface. The standard for sowing plants of particular species was (in kg·ha$^{-1}$): millet 20, buckwheat 70, white mustard 15, bird's-foot 35, Egyptian clover 20. In each year of research, garlic cloves were planted into a mass of green cover plants to the depth of 6 cm, previously determining the rows spaced 30 cm apart. Garlic planting was carried out on the following dates: 20 September 2013, 22 September 2014 and 28 September 2015.

In each year of research, garlic was harvested once in the vegetative phase, just before the appearance of generative shoots: 6 May 2014, 5 May 2015 and 2 May 2016. During garlic harvest, measurements of height (cm) and number of leaves per plant were carried out on 50 randomly selected plants, then the total yield and marketable yield were determined from 1 m$^2$, as well as the share of commercial yield in the total yield. Samples of garlic edible parts were also collected for chemical analyses.

### 2.3. Chemical Analyses

Phytochemical tests were carried out immediately after harvesting plants from the field in fresh plant material in the usable parts of garlic (bulb and leaf blade). The plant material was ground immediately before chemical analyses. Garlic bulbs and leaves were subject to the following determinations: dry matter by drying to a constant weight at 105 °C [36]; total and reducing sugars by the Schoorl–Luff method [37]; L-ascorbic acid by Tillman method with Pijanowski modification [38]. In garlic bulbs, the total ash was also analyzed [39].

In garlic leaves, the following parameters were determined: crude fiber, applying the Henneberg and Stochmann method [40]; total phenolic acids calculated as caffeic acid using the Arnova reagent [41]; essential oil by indirect steam distillation with the addition of m-xylene in Deryng apparatus [42].

### 2.4. Statistical Calculations

Cohran's test was used to test for homoscedasticity, following which, the data were subjected to a two-way analysis of variance (ANOVA), based on a factorial combination of mulch type × year. Means were separated by the least significance difference (LSD) test, when the *F*-test was significant. Data were evaluated by HSD Tukey test at $p < 0.05$. All calculations and analyses were performed using Statistica 10.0 PL software (StatSof Inc., Tulsa, OK, USA).

### 3. Results

The course of weather during the experiment differed from the average conditions in the region (Table 1). All the analyzed growing seasons of winter garlic were characterized by higher air temperature compared to the long-term average. Exceptionally warm were the 2014/2015 and 2013/2014 seasons, when in the period from September to April the average air temperature was higher than the long-term average of 1951–2010 by 2.7 and 2.5 °C, respectively. Equally warm was the 2015/2016 season, in which the average monthly air temperature was 2.0 °C higher than the long-term average. Winter and spring in 2015/2016 were characterized by an exceptionally high amount of rainfall; in the period from September to October they were 175 mm higher than the average multi-annual total for this period. The total precipitation in the same period in 2013/2014 was lower by 25 mm, and in 2013/2014 higher by 62 mm than the average multi-year sum. Plants cultivated with garlic as companion crops produced an abundant green mass in autumn, which, when left on the soil for the winter, was frozen, forming a layer of plant cover, protecting the soil surface from the damaging effects of external factors.

**Table 1.** Total rainfall and average air temperatures during the experimental trials (2013–2016), as compared to the long-term period (data from the Meteorological Station in Felin, 51.13º N; 22.37º E).

| Years | Month | | | | | | | | Average |
|---|---|---|---|---|---|---|---|---|---|
| | IX | X | XI | XII | I | II | III | IV | |
| Temperature (°C) | | | | | | | | | |
| 2013/2014 | 11.8 | 10.0 | 5.4 | 1.5 | −2.6 | 1.4 | 6.1 | 9.8 | 5.4 |
| 2014/2015 | 14.0 | 9.6 | 4.4 | 1.5 | 1.0 | 0.9 | 4.6 | 8.6 | 5.6 |
| 2015/2016 | 15.4 | 7.1 | 5.4 | 3.9 | −0.9 | −4.5 | 2.4 | 10.2 | 4.9 |
| Average 1951–2010 | 12.6 | 7.6 | 2.6 | −1.6 | −3.7 | −2.8 | 1.0 | 7.4 | 2.9 |
| Rainfall (mm) | | | | | | | | | Total |
| 2013/2014 | 65 | 5 | 80 | 16 | 71 | 12 | 48 | 45 | 342 |
| 2014/2015 | 25 | 26 | 23 | 30 | 47 | 7 | 56 | 41 | 255 |
| 2015/2016 | 113 | 50 | 52 | 31 | 41 | 62 | 63 | 44 | 456 |
| Average 1951–2010 | 54 | 40 | 38 | 31 | 23 | 26 | 28 | 39 | 279 |

Total and commercial yield, as well as the average plant height and number of leaves per plant in winter garlic cultivation per bunch harvest, are shown in Table 3. The average total yield of garlic was 4.5 kg·m$^{-2}$, and the commercial yield 3.3 kg·m$^{-2}$. There were no significant differences in yield, plant height and number of leaves per plant among years (Tables 2 and 3). Catch crop left until spring in the form of organic mulch had a significant impact on the total yield and commercial garlic yield for bunch harvest. The total yield of buckwheat, Egyptian clover and white mustard plants averaged from 5.3 to 5.8 kg·m$^{-2}$, which was more than twice as high as in the control plot (2.8 kg·m$^{-2}$). In addition, a higher total garlic yield was obtained with bird's-foot (by 50% on average) compared to the control, and the smallest with millet (3.1 kg·m$^{-2}$). Similarly, the highest commercial yield was in the variant with buckwheat, Egyptian clover and white mustard (from 3.9 to 4.3 kg·m$^{-2}$) than with bird's-foot (3.3 kg·m$^{-2}$), and the lowest with millet (2.3 kg·m$^{-2}$).

**Table 2.** Summary of statistical significance from analysis of variance.

| | Source of Variation | | |
| --- | --- | --- | --- |
| **Variable** | **Type of Mulch (TM)** | **Year (Y)** | **(TM) × (Y)** |
| Total yield | * | NS | * |
| Marketable yield | * | NS | * |
| Share of marketable yield in total yield | * | NS | NS |
| Plant height | * | NS | * |
| Number of leaves plant$^{-1}$ | * | NS | * |
| Bulb dry matter | NS | * | NS |
| Leaves dry matter | NS | * | NS |
| Bulb total sugar | * | * | NS |
| Leaves total sugar | * | * | * |
| Bulb reducing sugars | * | * | * |
| Leaves reducing sugars | * | * | * |
| Bulb L-ascorbic acid | * | * | NS |
| Leaves L-ascorbic acid | * | * | NS |
| Bulb total ash | * | ** | NS |
| Leaves crude fibre | * | * | * |
| Leaves phenolic acids | * | NS | * |
| Leaves essential oil | * | NS | * |

\* and \*\* indicate significant at *p* < 0.05 and 0.01, respectively. NS = not significant.

**Table 3.** Summary of statistical significance from analysis of variance.

| Treatments | Total Yield (kg·m$^{-2}$) | Marketable Yield (kg·m$^{-2}$) | Share of Marketable Yield in Total Yield (%) | Plant Height (cm) | Number of Leaves Plant$^{-1}$ |
| --- | --- | --- | --- | --- | --- |
| Type of mulch | | | | | |
| Millet | 3.10 ± 0.50 c | 2.33 ± 0.37 c | 75.2 ± 5.1 b | 29.1 ± 1.5 a | 4.1 ± 0.4 c |
| Buckwheat | 5.32 ± 0.53 a | 3.94 ± 0.39 a | 74.1 ± 3.4 c | 26.9 ± 1.8 ab | 4.7 ± 0.4 b |
| Mustard | 5.77 ± 0.41 a | 4.27 ± 0.31 a | 74.0 ± 4.1 c | 23.4 ± 2.7 c | 4.7 ± 0.1 b |
| Bird's-foot | 4.31 ± 0.21 b | 3.34 ± 0.16 b | 77.5 ± 3.4 a | 25.0 ± 2.0 bc | 4.9 ± 0.2 ab |
| Clover | 5.75 ± 0.30 a | 4.26 ± 0.22 a | 74.1 ± 4.1 c | 24.4 ± 2.8 bc | 5.1 ± 0.2 a |
| Control | 2.79 ± 0.34 c | 2.15 ± 0.26 c | 77.1 ± 4.0 a | 18.0 ± 1.7 d | 3.4 ± 0.2 d |
| Year | | | | | |
| 2013/2014 | 4.43 ± 1.30 a | 3.32 ± 0.93 a | 74.9 ± 4.9 a | 24.2 ± 4.0 a | 4.4 ± 0.6 a |
| 2014/2015 | 4.45 ± 1.26 a | 3.34 ± 0.91 a | 75.1 ± 5.1 a | 24.4 ± 3.6 a | 4.5 ± 0.6 a |
| 2015/2016 | 4.64 ± 1.32 a | 3.48 ± 0.94 a | 75.0 ± 5.0 a | 24.8 ± 4.6 a | 4.5 ± 0.6 a |

Different letters within each column and main factor indicate significant difference (*p* < 0.05).

Garlic plants in companion planting with catch crops were characterized by higher average height (from 23.4 to 29.1 cm) than in cultivation without plant litter (18.0 cm). When grown with millet and buckwheat, garlic plants were larger than those grown with mustard, bird's-foot and clover. Garlic plants formed more leaves when grown with bird's-foot and clover (4.9 to 5.1 pcs plant$^{-1}$), than with white mustard and buckwheat (4.7 pcs plant$^{-1}$) and with millet (4.1 pcs plant$^{-1}$), and the least in cultivation without catch crops in the form of mulch (3.4 pcs plant$^{-1}$).

The analysis of chemical composition showed a significant year effect (Table 4). In 2016, bulbs and leaves were characterized by higher dry matter (23.61% and 13.50%, respectively) and bulbs contained more total sugars (16.95% FW) and reducing sugars (0.55% FW) than in 2014 and 2015. In 2016 and 2015, leaves accumulated more total sugars (on average, 3.64% and 3.60% FW, respectively) and more reducing sugars (on average from 3.3% to 2.8% FW, respectively) than in 2014. Organic litter did not significantly affect the dry matter in garlic bulbs and leaves, because the differences were not significant compared to the control.

**Table 4.** Yield and plant biometric traits of winter garlic crop as affected by 'mulch type × year' interaction.

| Treatments | Total Yield (kg·m$^{-2}$) | Marketable Yield (kg·m$^{-2}$) | Plant Height (cm) | Number of Leaves Plant$^{-1}$ |
|---|---|---|---|---|
| Millet | | | | |
| 2013/2014 | 3.27 ± 0.45 ef | 2.45 ± 0.34 e | 29.6 ± 1.0 a | 4.2 ± 0.1 ce |
| 2014/2015 | 2.74 ± 0.39 f | 2.06 ± 0.29 e | 28.6 ± 2.6 ab | 4.1 ± 0.6 de |
| 2015/2016 | 3.29 ± 0.59 ef | 2.48 ± 0.44 e | 29.3 ± 0.6 a | 4.1 ± 0.5 de |
| Buckwheat | | | | |
| 2013/2014 | 4.75 ± 0.40 bd | 3.52 ± 0.29 bd | 27.4 ± 1.0 ac | 4.4 ± 0.2 bd |
| 2014/2015 | 5.46 ± 0.14 ac | 4.04 ± 0.10 ad | 26.4 ± 1.0 ad | 5.2 ± 0.0 a |
| 2015/2016 | 5.74 ± 0.40 a | 4.25 ± 0.30 ab | 27.0 ± 3.2 ad | 4.6 ± 0.1 ad |
| Mustard | | | | |
| 2013/2014 | 5.94 ± 0.40 a | 4.40 ± 0.29 a | 21.3 ± 1.0 d | 4.7 ± 0.1 ad |
| 2014/2015 | 5.42 ± 0.32 ac | 4.01 ± 0.24 ad | 22.6 ± 1.5 cde | 4.8 ± 0.1 ad |
| 2015/2016 | 5.96 ± 0.39 a | 4.41 ± 0.28 a | 26.3 ± 2.6 ae | 4.8 ± 0.1 ad |
| Bird's-foot | | | | |
| 2013/2014 | 4.28 ± 0.17 de | 3.32 ± 0.13 d | 22.7 ± 1.1 be | 4.9 ± 0.2 ac |
| 2014/2015 | 4.30 ± 0.07 de | 3.33 ± 0.24 d | 25.7 ± 1.1 ad | 4.8 ± 0.1 ad |
| 2015/2016 | 4.36 ± 0.37 ce | 3.38 ± 0.28 cd | 26.7 ± 1.1 ad | 5.2 ± 0.1 a |
| Clover | | | | |
| 2013/2014 | 5.80 ± 0.23 a | 4.29 ± 0.17 ab | 25.6 ± 3.5 ad | 5.3 ± 0.2 a |
| 2014/2015 | 5.78 ± 0.38 a | 4.28 ± 0.28 ab | 24.9 ± 1.5 ad | 5.2 ± 0.2 a |
| 2015/2016 | 5.66 ± 0.39 ab | 4.19 ± 0.29 ac | 22.6 ± 3.2 ce | 5.0 ± 0.1 ab |
| Control | | | | |
| 2013/2014 | 2.56 ± 0.29 f | 1.97 ± 0.22 e | 18.8 ± 1.0 e | 3.2 ± 0.1 f |
| 2014/2015 | 3.01 ± 0.41 f | 2.32 ± 0.32 e | 18.4 ± 2.5 e | 3.4 ± 0.0 ef |
| 2015/2016 | 2.82 ± 0.22 f | 2.17 ± 0.17 e | 16.8 ± 1.0 e | 3.5 ± 0.4 ef |

Different letters within each column indicate significant statistical differences ($p < 0.05$).

The average total sugar content in garlic bulbs was on average 16.0% FW, and in leaves 3.6% FW. Content of total sugars in garlic bulbs grown with white mustard and clover biomass (16.4 and 16.5% FW, respectively) did not differ significantly from the amount determined in control plants (17.4% FW). On the other hand, lower than in the control total sugar content (12% on average) was determined in garlic bulbs grown with organic mulch from bird's-foot, buckwheat and millet (Table 5).

Catch crops modified the total sugar content in leaves to a higher extent than in garlic bulbs. The content of total sugars in garlic leaves in crops with vegetable mulch was lower by an average of 24%–33% compared to the control. Among the assessed catch crops, more total sugars were determined in the leaves of garlic grown with litter of clover, mustard, buckwheat and bird's-foot (3.5%–3.2% FW), while less in those grown with millet (3.1% FW). A significant impact of the type of mulch years of cultivation and interaction of the total sugar content in garlic leaves was found. In the years 2013–2016,

the average total sugar content was higher in garlic leaves in control plots, and lower in cultivation with millet and bird's foot mulches (Table 6). The average content of reducing sugars in garlic bulbs was 0.5%, and 2.9% FW in leaves. The biomass of catch crops reduced the content of reducing sugars in garlic bulbs by an average of 6% to 30%, compared to the control. Among the organic mulches used to cover the soil, a higher level of reducing sugars was found in bulbs cultivated with mustard and clover (0.59% and 0.60% FW, respectively), and the least with bird's-foot, buckwheat and millet (on average 0.44% FW). The content of reducing sugars in leaves with biomass of catch crops from bird's-foot and clover did not differ significantly from the control. The smallest concentration of reducing sugars was found in garlic leaves in the cultivation with millet (2.3% FW).

**Table 5.** Effect of mulch type and year of cultivation on the content of dry matter, total and reducing sugars in winter garlic edible parts.

| Treatments | Dry Matter (%) | | Total Sugar (% FW) | | Reducing Sugars (% FW) | |
|---|---|---|---|---|---|---|
| | **Bulb** | **Leaves** | **Bulb** | **Leaves** | **Bulb** | **Leaves** |
| Type of mulch | | | | | | |
| Millet | 22.48 ± 1.46 a | 13.55 ± 0.38 a | 15.33 ± 0.90 b | 3.15 ± 0.02 e | 0.44 ± 0.02 c | 2.27 ± 0.17 c |
| Buckwheat | 22.53 ± 1.75 a | 12.28 ± 1.37 a | 15.22 ± 1.10 b | 3.45 ± 0.02 c | 0.44 ± 0.03 c | 2.75 ± 0.17 b |
| Mustard | 22.15 ± 1.50 a | 11.93 ± 1.71 a | 16.36 ± 1.10 a | 3.51 ± 0.04 bc | 0.59 ± 0.04 b | 2.76 ± 0.13 b |
| Bird's-foot | 22.32 ± 1.46 a | 12.51 ± 1.75 a | 15.41 ± 0.90 b | 3.24 ± 0.03 d | 0.44 ± 0.02 c | 2.82 ± 0.18 ab |
| Clover | 22.24 ± 1.40 a | 12.75 ± 1.74 a | 16.48 ± 1.00 a | 3.56 ± 0.06 b | 0.60 ± 0.03 b | 3.29 ± 0.22 a |
| Control | 22.21 ± 1.06 a | 12.85 ± 0.53 a | 17.39 ± 0.80 a | 4.68 ± 0.16 a | 0.64 ± 0.03 a | 3.30 ± 0.20 a |
| Year | | | | | | |
| 2013/2014 | 21.45 ± 1.11 b | 11.97 ± 1.26 b | 15.41 ± 1.10 b | 3.55 ± 0.46 b | 0.50 ± 0.09 b | 2.76 ± 0.68 b |
| 2014/2015 | 21.91 ± 0.93 b | 12.46 ± 1.48 b | 15.74 ± 1.00 b | 3.60 ± 0.53 a | 0.52 ± 0.09 b | 2.81 ± 0.69 a |
| 2015/2016 | 23.61 ± 1.09 a | 13.50 ± 1.04 a | 16.95 ± 1.00 a | 3.64 ± 0.56 a | 0.55 ± 0.09 a | 3.03 ± 0.62 a |

Different letters within each column and main factor indicate significant difference ($p < 0.05$).

**Table 6.** Chemical composition of winter garlic edible parts as affected by 'mulch type × year' interaction.

| Treatments | Leaves Total Sugar (% FW) | Leaves Crude Fibre (% FW) | Leaves Phenolic Acids (mg·g⁻¹ FW) | Leaves Essential Oil (% FW) | Reducing Sugars (% FW) | |
|---|---|---|---|---|---|---|
| | | | | | **Bulb** | **Leaves** |
| Millet | | | | | | |
| 2013/2014 | 3.1 ± 0.0 d | 1.12 ± 0.01 c | 0.32 ± 0.01 ab | 0.30 ± 0.06 a | 0.44 ± 0.00 c | 2.0 ± 0.0 d |
| 2014/2015 | 3.1 ± 0.0 d | 1.13 ± 0.00 c | 0.35 ± 0.01 a | 0.34 ± 0.00 a | 0.41 ± 0.00 c | 2.2 ± 0.0 c |
| 2015/2016 | 3.1 ± 0.0 d | 1.16 ± 0.01 c | 0.38 ± 0.00 a | 0.34 ± 0.00 a | 0.47 ± 0.02 c | 2.4 ± 0.0 c |
| Buckwheat | | | | | | |
| 2013/2014 | 3.4 ± 0.0 bc | 0.76 ± 0.01 de | 0.29 ± 0.00 bc | 0.29 ± 0.00 ab | 0.40 ± 0.00 c | 2.7 ± 0.0 b |
| 2014/2015 | 3.4 ± 0.0 b | 0.88 ± 0.02 d | 0.29 ± 0.00 bc | 0.29 ± 0.00 ab | 0.45 ± 0.01 c | 2.5 ± 0.0 c |
| 2015/2016 | 3.4 ± 0.0 b | 0.92 ± 0.05 d | 0.21 ± 0.03 cd | 0.29 ± 0.00 ab | 0.48 ± 0.00 c | 2.9 ± 0.1 b |
| Mustard | | | | | | |
| 2013/2014 | 3.5 ± 0.0 b | 0.72 ± 0.01 ef | 0.30 ± 0.04 ab | 0.31 ± 0.00 ab | 0.59 ± 0.05 b | 2.6 ± 0.0 c |
| 2014/2015 | 3.5 ± 0.0 b | 0.59 ± 0.04 f | 0.36 ± 0.01 a | 0.31 ± 0.00 ab | 0.59 ± 0.02 b | 2.7 ± 0.0 bc |
| 2015/2016 | 3.5 ± 0.0 b | 0.66 ± 0.03 ef | 0.30 ± 0.01 ab | 0.31 ± 0.00 ab | 0.60 ± 0.05 ab | 2.9 ± 0.0 b |
| Bird's-foot | | | | | | |
| 2013/2014 | 3.2 ± 0.0 cd | 0.88 ± 0.01 d | 0.33 ± 0.02 ab | 0.32 ± 0.00 ab | 0.41 ± 0.01 c | 2.6 ± 0.1 c |
| 2014/2015 | 3.2 ± 0.0 d | 0.98 ± 0.01 d | 0.28 ± 0.02 bcd | 0.32 ± 0.00 ab | 0.43 ± 0.02 c | 2.7 ± 0.1 b |
| 2015/2016 | 3.2 ± 0.0 cd | 0.92 ± 0.04 d | 0.31 ± 0.08 ab | 0.32 ± 0.00 ab | 0.47 ± 0.01 c | 3.0 ± 0.0 b |
| Clover | | | | | | |
| 2013/2014 | 3.5 ± 0.0 b | 0.59 ± 0.01 f | 0.27 ± 0.02 bcd | 0.27 ± 0.00 bc | 0.59 ± 0.02 b | 3.2 ± 0.2 a |
| 2014/2015 | 3.5 ± 0.0 b | 0.71 ± 0.06 ef | 0.32 ± 0.02 ab | 0.27 ± 0.00 bc | 0.59 ± 0.02 b | 3.2 ± 0.1 a |
| 2015/2016 | 3.6 ± 0.0 b | 0.61 ± 0.05 f | 0.34 ± 0.08 a | 0.27 ± 0.00 bc | 0.60 ± 0.02 ab | 3.3 ± 0.3 a |
| Control | | | | | | |
| 2013/2014 | 4.5 ± 0.0 a | 1.57 ± 0.01 b | 0.19 ± 0.02 d | 0.22 ± 0.00 c | 0.61 ± 0.01 ab | 3.1 ± 0.1 a |
| 2014/2015 | 4.7 ± 0.1 a | 1.61 ± 0.04 b | 0.20 ± 0.01 cd | 0.22 ± 0.00 c | 0.63 ± 0.01 ab | 3.2 ± 0.1 a |
| 2015/2016 | 4.8 ± 0.1 a | 1.87 ± 0.10 a | 0.24 ± 0.00 bcd | 0.22 ± 0.00 c | 0.68 ± 0.00 a | 3.5 ± 0.1 a |

Different letters within each column indicate significant statistical differences ($p < 0.05$).

In all years of research, smaller amounts of reducing sugars were found in bulbs of garlic cultivated with bird's foot, millet and buckwheat mulch, and more on control, and in 2016, a year with high rainfall, however only cultivations with clover and buckwheat. In the 2013–2016 research years, more reducing sugars were found in garlic leaves in cultivation with clover and without soil mulching (control). Fewer reducing sugars were accumulated in garlic leaves in the cultivation with millet, mustard and bird's-foot in 2014, when the average temperature in February–April significantly exceeded the long-term average. A low amount of reducing sugars was reported in garlic leaves in the cultivation of buckwheat in 2015, a year with low rainfall for the vegetation period.

In the dry years of 2014 and 2015, the lowest content of L-ascorbic acid was found in garlic leaves and bulbs, and the lowest level of raw fiber and total ash in leaves, while the highest content was observed in the wet 2016 (Table 1). There was no significant difference in the content of phenolic acids and essential oil in garlic leaves among the years of research. The content of L-ascorbic acid in bulbs was on average 11.4 mg·100 g$^{-1}$ and in leaves 84.1 mg·100 g$^{-1}$ FW (Table 7). The concentration of L-ascorbic acid in bulbs and leaves of garlic grown with mustard and clover did not differ significantly from the amount of this component from the control. The biomass from catch crops from bird's-foot, buckwheat and millet reduced the content of L-ascorbic acid in bulbs and garlic leaves by an average of 10% compared to the control.

**Table 7.** Effect of mulch type and year of cultivation on chemical composition of winter garlic edible parts.

| Treatments | L-ascorbic Acid (mg·100 g$^{-1}$ FW) | | Total Ash in Bulb (% DM) | Crude Fibre in Leaves (% FW) | Phenolic Acids in Leaves (mg·g$^{-1}$ FW) | Essential Oil in Leaves (% FW) |
|---|---|---|---|---|---|---|
| | **Bulb** | **Leaves** | | | | |
| Type of mulch | | | | | | |
| Millet | 10.7 ± 0.7 b | 79.7 ± 5.1 b | 0.87 ± 0.07 b | 1.13 ± 0.01 b | 0.35 ± 0.02 a | 0.32 ± 0.04 a |
| Buckwheat | 10.8 ± 0.8 b | 79.8 ± 6.2 b | 0.96 ± 0.03 a | 0.85 ± 0.08 d | 0.26 ± 0.03 b | 0.29 ± 0.01 bc |
| Mustard | 11.9 ± 0.8 a | 88.3 ± 6.0 a | 0.96 ± 0.03 a | 0.66 ± 0.06 e | 0.32 ± 0.03 a | 0.31 ± 0.02 ab |
| Bird's-foot | 10.7 ± 0.7 b | 79.1 ± 5.1 b | 0.97 ± 0.02 a | 0.92 ± 0.04 c | 0.31 ± 0.05 a | 0.32 ± 0.00 a |
| Clover | 12.0 ± 0.7 a | 88.7 ± 5.6 a | 0.97 ± 0.02 a | 0.63 ± 0.06 e | 0.31 ± 0.04 a | 0.27 ± 0.00 c |
| Control | 11.9 ± 0.5 a | 88.6 ± 4.2 a | 0.97 ± 0.01 a | 1.68 ± 0.15 a | 0.21 ± 0.02 c | 0.22 ± 0.01 d |
| Year | | | | | | |
| 2013/2014 | 10.9 ± 0.8 b | 80.8 ± 6.6 b | 0.94 ± 0.03 b | 0.94 ± 0.33 c | 0.28 ± 0.04 a | 0.28 ± 0.04 a |
| 2014/2015 | 11.1 ± 0.8 b | 82.5 ± 6.1 b | 0.94 ± 0.07 b | 0.98 ± 0.31 b | 0.30 ± 0.05 a | 0.29 ± 0.04 a |
| 2015/2016 | 12.0 ± 0.7 a | 88.8 ± 5.3 a | 0.97 ± 0.02 a | 1.02 ± 0.43 a | 0.30 ± 0.06 a | 0.29 ± 0.04 a |

Different letters within each column and main factor indicate significant differences ($p < 0.05$).

The content of crude fiber, phenolic acids and essential oil was determined in garlic leaves. The biomass of catch crops reduced the content of crude fiber in leaves by an average of 33%–63%, in comparison with the control. The least crude fiber was found in garlic leaves with litter of clover (0.63%) and mustard (0.66%).

The concentration of total ash in garlic bulbs in cultivation with biomass from catch crops, except for millet, did not differ significantly from the amount determined in the bulbs of plants harvested from the control. Significantly less total ash was determined in garlic bulbs cultivated with millet biomass (0.87% DM) compared to the control and other mulch types. The interaction of weather conditions with the content of total ash in garlic bulbs was shown. The highest crude fiber content was recorded in garlic leaves harvested in 2016, from both the control and catch crops of clover, bird's-foot, mustard and buckwheat; the lowest in garlic bulbs harvested in 2014 and 2015 with millet cultivation.

The content of phenolic acids was, on average, 0.29 mg·g$^{-1}$ FW. Plants cultivated with the biomass of catch crops contained more phenolic acids (on average from 19% to 66% more) than from the control. The highest concentration of phenolic acids was found in leaves from garlic plants cultivated with the biomass of catch crops from millet, clover, bird's-foot and mustard (0.31–0.35 mg·g$^{-1}$ FW, on average), the smallest from buckwheat (0.26 mg·g$^{-1}$ FW).

The interaction of the factors examined shows that in the years 2013–2016 the highest content of phenolic acids was found in the leaves of garlic grown under litter of mustard and millet, as well as that of clover (in 2015 and 2016) and bird's-foot (in 2014 and 2016). Garlic leaves with buckwheat litter in 2016, bird's-foot in 2015 and clover in 2014 had a lower content of phenolic acids. On average, 0.28% of essential oil was determined in garlic leaves. Plants cultivated with biomass of catch crops accumulated on average by 23%–45% more essential oil in the leaves than in the control plot. More essential oil was determined in the leaves of plants grown with mustard, bird's-foot and millet (0.31%–0.32%); the least with buckwheat biomass (0.29%).

Analyzing the effect of interaction between litter type of mulch and year of research, a relatively stable level of oil in the leaves was found in cultivation with organic litter. In 2013–2016, more essential oil was found in garlic leaves when grown with millet litter, and less when grown with that of clover.

## 4. Discussion

Our results indicated that the biomass of catch crops in the form of mulch significantly increased the total yield from 1.5 to 2.9 kg·m$^{-2}$ on average, and the commercial yield of garlic from 1.1 to 2.1 kg·m$^{-2}$ compared to the control. These results confirmed previous findings on the yield-forming effect of organic litter in winter garlic [5,19,43]. However, garlic plants reacted differently to the presence of spring catch crops. The best crop-forming effects were obtained by using clover, mustard and buckwheat as catch crops. It can be assumed that these differences may be associated with the different rates of decomposition of the plant mass into minerals that may be available to garlic plants. In cultivation of the garlic winter (cv. 'Harnaś'), a higher marketable yield (4.3 kg·m$^{-2}$) was obtained during the July harvest, two months later, as in this experiment condition [17].

According to Hayden et al. [44] and Tribouillois et al. [45], an effective cover crop accumulates most of the nutrients in the summer-autumn period. The collected minerals and catch crops built into the biomass are released in spring in soil activity processes, due to which they can be absorbed by the plants [35]. Kosterna [22], Ngouajio and Mennan [46], Sinkevičiene et al. [47], and Guo et al. [48] reported on the significant relationship between the type of catch crop used for companion planting and the increase in yield of plant species cultivated in the main crop.

Roper et al. [49] showed that the yield-forming effects of green fertilizers in the first year after their application depends on spring weather conditions. Heat and moisture are necessary for the mineralization of organic matter. According to Vincent-Caboud et al. [30] and Shackelford et al. [14], cool and dry spring inhibits mineralization, while warm and wet one accelerates it. The obtained test results confirmed this relationship by lack of significant differences in the total yield and commercial garlic yield among the years. In early spring, atmospheric conditions were similar and the soil showed high biological activity. In 2014, 2015 and 2016, air temperatures were already positive in March and exceeded the long-term average values by 5.1, 3.6 and 1.4 °C, respectively, while the amount of precipitation was higher than the long-term average by 20, 28 and 35 mm, respectively.

Under experimental conditions, garlic plants in companion cultivation with catch crops were of higher height and developed more leaves than in the cultivation of homogeneous garlic on the control. On average, garlic plants on plots with clover and bird's-foot developed more leaves. These differences could have been caused by the greater availability of mineral substances on plots with organic litter, as well as atmospheric conditions, as evidenced by the importance of interaction. Organic matter contains a lot of nitrogen; the faster it is decomposed, the sooner nutrients are available to plants [35].

The biomass of catch crops affects not only the size but also the quality of the crop, and thus also the content of chemical components found in bulbs and leaves of winter garlic in cultivation for bunch harvesting. In this work, the average dry matter content of bulbs was 22.3%, and of leaves 12.6%. The use of biomass for catch crops in the form of mulch had no effect on the dry matter content of bulbs and garlic leaves. Similarly, in the research of Faradonbeh et al. [5], the dry matter content of garlic bulbs fertilized with biomass rice straw, compared to cultivation without straw, did not differ

significantly. More dry matter (14.2%–14.7%), but only in the leaves, was determined in the cultivation of plants covered with perforated foil and polypropylene nonwoven [7].

The chemical composition of garlic plants was largely affected by weather conditions. A significantly higher content of dry matter and L-ascorbic acid in garlic bulbs and leaves was recorded in 2016, a year with moderate rainfall and high temperatures in April (10.2 °C).

In garlic plants grown for bunch harvesting intended for direct consumption, major importance is attached to total sugars, and in those intended for processing it is the reducing sugars. In our research, bulbs of garlic grown with bird's-foot, mustard and millet contained less total sugars (12% on average) compared to the control. All catch crops under study reduced the content of total sugars in leaves (by 25%–33% on average), as well as that of reducing sugars (by 30% on average), in comparison to those found in cultivation without catch crops. In comparison to the control, it was only the cultivation of winter garlic with bean plants, i.e., with clover, that did not reduce the total sugar content in bulbs and winter garlic cultivation with clover, and bird's-foot that did not affect the content of reducing sugars in garlic leaves. This is in line with the results of other authors. According to Danilchenko et al. [50], straw mulches with high carbon content reduce the concentration of sugars in potato tubers, similarly to organic litter from corn and buckwheat rye straw, which seem to reduce the total sugar content in tomato fruits and the edible parts of broccoli [22]. In contrast, the biomass of catch crops from bean plants, with a favorable quantitative carbon to nitrogen ratio, increases the content of sugar in potato tubers [51].

The sugar content in garlic plants was strongly affected by the weather. In 2016, bulbs contained more total sugars, reducing sugars and total ash compared to 2014 and 2015. In 2016, in February and March, the rainfall was abundant, while the temperature was low. Therefore it can be assumed that the loss of mineral components from the soil profile was lower.

In this experiment, the average amount of L-ascorbic acid in bulbs was 11.4 mg·100 g$^{-1}$ FW, and in leaves 84.1 mg·100 g$^{-1}$ FW. Bulbs and leaves contained more L-ascorbic acid (14 and 100 mg·100 g$^{-1}$ FW, respectively) in plants covered with foil and non-woven fabric [7]. In our research, the impact of various kinds of organic litter on L-ascorbic acid content was particularly interesting. The accumulation of L-ascorbic acid in garlic bulbs and leaves proved to be promoted by mulching the soil with clover and mustard, while it was decreased by mulching with bird's-foot, buckwheat and millet. Antioxidants such as L-ascorbic acid were found to increase or decrease in response to environmental stress [32]. It can be assumed that the biomass produced limited, in various ways, the heating of the soil, thereby reducing evaporation and promoting water absorption and infiltration. Similar results were obtained by Najafabadi et al. [19], who reported that in a dry climate, mulching the soil with straw increases the amount of L-ascorbic acid in garlic bulbs, and in a moderate climate, it shows the same effect in sweet pepper fruit [20] and in the edible parts of tomato and broccoli [22]. In our experiment, the crude fiber content was 0.98%. More crude fiber was definitely determined in the leaves of control plants grown without plant cover. The content of crude fiber depends on many factors, including thermal and humidity conditions, as evidenced by the significance of interaction between the type of litter and year. It can be assumed that organic mulch may have reduced mineral leaching and slowed down the mineralization process. The least crude fiber was determined in the leaves of plants grown together with clover and mustard cover, and the most in cultivations with millet. Hayden et al. [44] expressed the view that large amounts of organic matter ensure the optimal circulation of minerals and create conditions for plants to use them more slowly and thus produce lower amounts of crude fiber than when grown on uncovered soil.

From the point of view of the consumption value of garlic grown for bunch harvesting, an important issue is the content of phenolic acids and essential oil in leaves [52]. The content of phenolic acids is highly variable, depending on cultivar, growth conditions and fertilization [53]. Organic fertilizers have been found to increase the content of polyphenolic compounds in cultivations of bean [54], beetroot [55] and tomato [56,57]. This is consistent with the results of our research. In the experiment conducted, the leaves contained more phenolic acids in the cultivation of garlic with the biomass of

cover plants than on the control. The content of oil in garlic leaves was modified by the type of organic mulch. A higher content of essential oil was characteristic for leaves cultivated with millet, bird's-foot, and mustard, and lower with clover and buckwheat. Millet, bird's-foot and mustard plants formed lush and carbon-rich biomass, contributing more to humus. The impact of climatic conditions on the collection and composition of the oil also seems likely, as indicated by divergent results of research conducted in different parts of the world [4]. Such an impact was not confirmed by our research, because differences in the amount of oil leaves between the years of research were not significant. This shows that the plant mulches used in the study favoured the stable accumulation of essential oil in garlic leaves, regardless of the changing weather conditions in the years of research.

## 5. Conclusions

The three-year study showed the usefulness of spring catch crops (millet, buckwheat, white mustard, bird's-foot and Egyptian clover) for mulching the soil surface in the cultivation of winter garlic. Indeed, the catch crops in spring formed a natural organic mulch, which did not decrease the commercial yield of garlic than the conventional cultivation, i.e., without plant mulching. The chemical composition of garlic plants grown with catch crops differed from those conventionally grown. Particularly, catch crops such as clover and mustard increased the content of L-ascorbic acid in garlic, as well as the content of phenolic acids and essential oil. By contrast, the content of crude fiber was lower in the edible parts of garlic grown with catch crops, as well as generally that of total sugars and reducing sugars in garlic leaves and bulbs. Although certain seasonal variations were here observed on the impact of catch crops on the garlic productive and qualitative traits, it can be assumed that the introduction of catch crops in the form of natural mulch for the cultivation of winter garlic may enhance some chemical traits of crop edible parts, while not affecting the commercial yield. Further research is required, to address the impact of cover crops on the microclimate and experiments conducted in different agro-environmental conditions which are needed to obtain more practically oriented conclusions for vegetable growers. From a practical view, this should be researched further by combining different cover crop plants with different main crop species and varieties, to avoid problems related to plant competition.

**Author Contributions:** H.B. and A.S. assumed the idea, conducted the experiment and implemented the measurements. A.S., G.P. and S.L. wrote the manuscript; G.P. and S.L. revised the manuscript. All authors have read and agreed to the published version of the manuscript.

**Funding:** The research received no external funding.

**Acknowledgments:** The authors appreciate the Małopolska Plant Breeding Agency in Zamość for assistance in conducting the field experiment.

**Conflicts of Interest:** The authors declare no conflict of interest.

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
