# Peer review of "Influence of Catch Crops on Yield and Chemical Composition of Winter Garlic Grown for Bunch Harvesting"

_agriculture, doi:10.3390/agriculture10040134_

Round 1

Reviewer 1 Report

Summary

The research developed in this article is very interesting. Five different types of mulch are evaluated during three agronomic cycles, these crops are used as mulch (after freezing) in winter garlic crops. They evaluate the influence on the production and chemical quality parameters of garlic. The results are highly relevant and consistent, especially the increase in crop yield that is twice as high as the control treatment with some of the plant species tested.

Broad comments

Regarding the chemical parameters measured, I am surprised that they are expressed as a percentage by fresh weight instead of a percentage by dry weight. Although there are authors who use percentages in fresh weight, it is more common to refer to them in dry weight. I consider it to be preferable in dry weight, mainly because the moisture content of plant tissues can vary due to numerous factors, and this variation is much more noticeable in non-irrigated crops such as garlic, especially when there is great variability in yearly rainfall or when there are different covering materials or not in the soil, due to their influence on evaporation. I usually find in my field research great differences in fresh weights in vegetables grown with drip irrigation (dry weights show less variability), in crops without irrigation the variability should be greater.

This consideration may be the reason why there are no significant differences of the type of mulch in parameters such as dry matter or sugars and if there is a significant influence of the year factor, the more or less rainy years can affect the fresh weight of the bulb and the leaves at the time of harvest.

I consider than tables of chemical parameters and statistical analysis should be performed on values on dry matter better than on values on fresh matter if this is possible.

The discussion of the performance parameters of the trial should be carried out considering the results of the previous work carried out in the same area by two of the same authors.

SaÅ‚ata, A., Moreno-Ramon, H., Ibáñez-Asensio, S., Buczkowska, H., NurzyÅ„ska-Wierdak, R., Witorożec, A., Parzymies, M. (2017). Possibilities to improve soil physical properties in garlic cultivation with cover crops as living mulches.Acta Sci. Pol. Hortorum Cultus, 16(6), 157–166. DOI: 10.24326/asphc.2017.6.14

It would be enriching to attribute possible causes to the notable differences in crop yield between the two articles.

Specific comments

Line 111 were instead of wreLine 118-126 The fertilization used in the control plots must also be indicated.Line 173 Table 3 referenced does not appear in the document. A table 4 appears after table 2, indicating that an error has occurred in the numbering of the tables that affects tables 4, 5, 6 and 7.

Author Response

Q: Regarding the chemical parameters measured, I am surprised that they are expressed as a percentage by fresh weight instead of a percentage by dry weight. Although there are authors who use percentages in fresh weight, it is more common to refer to them in dry weight. I consider it to be preferable in dry weight, mainly because the moisture content of plant tissues can vary due to numerous factors, and this variation is much more noticeable in non-irrigated crops such as garlic, especially when there is great variability in yearly rainfall or when there are different covering materials or not in the soil, due to their influence on evaporation. I usually find in my field research great differences in fresh weights in vegetables grown with drip irrigation (dry weights show less variability), in crops without irrigation the variability should be greater. This consideration may be the reason why there are no significant differences of the type of mulch in parameters such as dry matter or sugars and if there is a significant influence of the year factor, the more or less rainy years can affect the fresh weight of the bulb and the leaves at the time of harvest. I consider than tables of chemical parameters and statistical analysis should be performed on values on dry matter better than on values on fresh matter if this is possible.

A: the aim of this study was to evaluate the influence of some agricultural practices on the yield and chemical composition of garlic cultivated for bunch harvest with green leaves.

In several works the cultivation of vegetables for the purpose of harvest for fresh consumption, the chemical composition of vegetables is expressed as fresh weight (see some references given below). In addition, the dry matter content is reported, so that readers could refer the values of chemical parameters as dry weight.

  1. Rekowska, E.; Skupień, K. The influence of selected agronomic practices on the yield and chemical composition of winter garlic. Crops Res. Bull. 2009, 70, 173-182.
  2. Kosterna E. The effect of soil mulching with straw on the yield and selected components of nutritive value in broccoli and tomatoes. Folia Hort. 2014, 26/1, 31-42
  3. Najafabadi Mahdieh, M.B.; Peyvast, Gh.; Hassanpour Asil, M.; Olfati, J.A.; Rabiee, M. Mulching effects on the yield and quality of garlic as second crop in rice fields. J. Plant Prod. 2012, 6, 3, 279-290.

Q: The discussion of the performance parameters of the trial should be carried out considering the results of the previous work carried out in the same area by two of the same authors.

SaÅ‚ata, A., Moreno-Ramon, H., Ibáñez-Asensio, S., Buczkowska, H., NurzyÅ„ska-Wierdak, R., Witorożec, A., Parzymies, M. (2017). Possibilities to improve soil physical properties in garlic cultivation with cover crops as living mulches.Acta Sci. Pol. Hortorum Cultus, 16(6), 157–166. DOI: 10.24326/asphc.2017.6.14

It would be enriching to attribute possible causes to the notable differences in crop yield between the two articles.

A: a comment was added in lines 293-294 to compare the results between the two manuscripts.

Q: Line 111 were instead of wre

A: done (line 132).

Q: Line 118-126 The fertilization used in the control plots must also be indicated.

A: provided in lines 126-127.

Q: Line 173 Table 3 referenced does not appear in the document. A table 4 appears after table 2, indicating that an error has occurred in the numbering of the tables that affects tables 4, 5, 6 and 7.

A: all these mistakes in the numbering of the tables were corrected in the revised manuscript.

Reviewer 2 Report

The three years-long research, aiming to study thei Influence of  different catch crops on yield and chemical composition of winter garlic for bunch harvesting, is very interesting.

Introduction given a good background of garlic agro-tecnique. Authors could describe the importance of garlic in Poland and so, why this experiment was made up in this area.

Material and Methods could be improved. In line 95-105, authors describe the weather conditions in Lublin, but in general...so I think that in this section you have to write what and how you measured during the experiment. Line 95-105 could be better in introduction.

There are some mistakes in the manuscript (es. line 111 wre) check it.

Conclusion are supported by results.

Author Response

Q: Introduction given a good background of garlic agro-tecnique. Authors could describe the importance of garlic in Poland and so, why this experiment was made up in this area.

A: provided in lines 39-43.

Q: In line 95-105, authors describe the weather conditions in Lublin, but in general...so I think that in this section you have to write what and how you measured during the experiment. Line 95-105 could be better in introduction.

A: in section 2.1 we described the general weather conditions in the area of experimental field with the aim to characterize the specific conditions in which the research was conducted. We believe that such information are best placed in the Materials and Methods. According to the Reviewer comments, we added information on what and how we measured the meteorological data during the studied growing seasons.

A: provided in lines 111-112

Q: There are some mistakes in the manuscript (es. line 111 wre) check it.

A: corrected in line 132.

Round 2

Reviewer 1 Report

I agree with the changes made by the authors in the manuscript

I still consider than tables of chemical parameters and statistical analysis should be performed on values on dry matter better than on values on fresh matter if this is possible. But if the marketable product are garlic cultivated for bunch harvest with green leaves, the results on fresh vallues are also quite interesting, I agree with autors.